# Deep Eutectic Solvents as New Reaction Media to Produce Alkyl-Glycosides Using Alpha-Amylase from *Thermotoga maritima*

**DOI:** 10.3390/ijms20215439

**Published:** 2019-10-31

**Authors:** Alfonso Miranda-Molina, Wendy Xolalpa, Simon Strompen, Rodrigo Arreola-Barroso, Leticia Olvera, Agustín López-Munguía, Edmundo Castillo, Gloria Saab-Rincon

**Affiliations:** Departamento Ingeniería Celular y Biocatálisis, Instituto de Biotecnología, Universidad Nacional Autónoma de México, Apartado Postal 510-3, Cuernavaca, Morelos 62250, Mexico; alfonso_itz@yahoo.com.mx (A.M.-M.); wxolalpa@ibt.unam.mx (W.X.); simon.strompen@basf.com (S.S.); rarreolb@ibt.unam.mx (R.A.-B.); lolvera@ibt.unam.mx (L.O.); agustin@ibt.unam.mx (A.L.-M.); edmundo@ibt.unam.mx (E.C.)

**Keywords:** Enzymatic glycosylation, alkyl glycosides (AG)s, Deep eutectic solvents (DES), Amy A, alcoholysis, hydrolysis, methanol, circular dichroism, protein stability, alpha-amylase

## Abstract

Deep Eutectic Solvents (DES) were investigated as new reaction media for the synthesis of alkyl glycosides catalyzed by the thermostable α-amylase from *Thermotoga maritima* Amy A. The enzyme was almost completely deactivated when assayed in a series of pure DES, but as cosolvents, DES containing alcohols, sugars, and amides as hydrogen-bond donors (HBD) performed best. A choline chloride:urea based DES was further characterized for the alcoholysis reaction using methanol as a nucleophile. As a cosolvent, this DES increased the hydrolytic and alcoholytic activity of the enzyme at low methanol concentrations, even when both activities drastically dropped when methanol concentration was increased. To explain this phenomenon, variable-temperature, circular dichroism characterization of the protein was conducted, finding that above 60 °C, Amy A underwent large conformational changes not observed in aqueous medium. Thus, 60 °C was set as the temperature limit to carry out alcoholysis reactions. Higher DES contents at this temperature had a detrimental but differential effect on hydrolysis and alcoholysis reactions, thus increasing the alcoholyisis/hydrolysis ratio. To the best of our knowledge, this is the first report on the effect of DES and temperature on an enzyme in which structural studies made it possible to establish the temperature limit for a thermostable enzyme in DES.

## 1. Introduction

Alkyl glycosides (AGs) are a class of non-ionic surfactants prepared from renewable agricultural resources, namely starch and fats or their derivatives [1]. AGs contain a carbohydrate head group such as glucose, galactose, maltose, xylose, α-cyclodextrin, and a hydrocarbon tail, usually a primary alcohol of different chain length of saturated or unsaturated nature [2,3]. Their high biodegradability and a low toxicity make them attractive for many applications such as cosmetics, foods, the extraction of organic dyes, membrane protein research, and pharmaceuticals [4]. The industrial production of alkyl glycosides has been typically achieved by traditional organic chemical procedures. However, this strategy requires multiple protection, deprotection and activation steps, the preparation of an anomerically-pure AG requiring high temperature and pressure, as well as the use of toxic catalysts, harmful chemicals, dangerous solvents, and other harsh conditions that adversely impact the environment and human health [4,5].

Considering that one of the most interesting properties of enzymes is their regio- and stereo- selectivity, a biocatalytic approach represents an interesting alternative to cope with this limitation. Moreover, the reactions are conducted under mild temperature and pH conditions which minimize side reactions [6,7,8,9,10,11,12,13]. In this context, AGs have been successfully prepared enzymatically, by applying glycosidases, such as *β*-galactosidases, *β*-glucosidases [14], or *β*-xylosidases [15], among others. Nevertheless, reports dealing with α-amylases are scarce [1,16,17,18]. 

The natural reaction of *α*-amylases (E.C. 3.2.1.1) is the hydrolysis of internal α-1,4-glycosidic bonds in starch through a double-displacement mechanism in which a covalent intermediate glycosyl enzyme is deglycosylated by water [19]. Like all retaining glycosidases, α-amylases can also catalyze transfer reactions which result from the affinity of alternative molecules other than water as glucosyl acceptors. Sugars are the most efficient group of acceptors studied (transglycosylation reactions), but glucosyl residue may also be transferred to alcohols (alcoholysis reactions) [20]. As water is the natural medium in these reactions, the transfer yield is always limited by the competition with hydrolysis. Deep eutectic solvents (DES) usually consist of a salt (most frequently quaternary ammonium salts such as choline chloride) and a complexing agent acting as a hydrogen bond donor (e.g. urea). Replacing the conventional aqueous solvent by alternative solvents such as DES offers the possibility to favor transglycosylation activity. Therefore, together with enzyme engineering, a strategy based on solvent engineering is a promising alternative to improve the synthesis yield of industrially-relevant alkyl-glycosides.

The aim of this work was to study the effect of DES as alternative reaction media for the α-amylase catalyzed synthesis of alkyl-glycosides. DES share many characteristics of ionic liquids, including their capacity to dissolve compounds of different polarities, their availability, and their environmental friendliness. Recent studies indicate that pure DESs or cosolvents represent “green” alternatives for reaction media in lipase-catalyzed synthesis [21,22,23]. Few reports exist, however, on enzymes reactions other than lipases in DES, as they generally have high viscosities and require temperatures higher than 50 °C, either for melting, or simply to reduce the medium viscosity. Amy A, an α-amylase from *T. maritima*, is a stable enzyme, active at temperatures above 80 °C, and thus considered particularly suited for application in DES. 

## 2. Results

In the present work, a series of 12 deep eutectic solvents (DES) were prepared. The selection of DES was based on viscosity and literature data reporting biocatalytic reactions in the corresponding solvents. Notably, natural deep eutectic solvents (NADES) without water or choline chloride as a salt are highly viscous, and therefore, are unsuitable for enzymatic reactions. On the other hand, Cholinium (Ch+), a natural Generally Recognized as Safe (GRAS) compound, commonly used as a food additive, is the preferred hydrogen bond acceptor (HBA) [24]. Thus, choline chloride was the most common HBA selected in combination with hydrogen bond donors (HBD) of different structure: amides, carboxylic acids, alcohols, and polyols (sugars). The composition of all the assayed DES and their preparation method are reported in Appendix A.

### 2.1. Screening of DES for Application in α-amylase Catalyzed Reactions and Effect of Water Content on Enzyme Activity and Stability

While lipases can be used in essentially water-free media or systems with low water activity [25,26], most enzymes require water for activity [27,28,29,30,31]. In this context, a screening in 2 stages was designed to evaluate the viability of DESs in Amy A catalyzed reactions. In a first stage, DESs were assayed as pure solvent and cosolvent in order to define the minimal amount of water required for the reaction. 

In the second stage, the DES or water: DES mixtures were assayed as reaction media for transfer reactions with methanol as the acceptor substrate, using maltotriose as a glucosyl donor.

To identify the required amount of water for enzyme activity, the 12 selected DES were used as solvents in reactions containing increasing water concentrations of 0, 5, 10, 25, 50, 75, and 100% (*v/v*), with maltotriose as a substrate. Substrate conversion was measured as the amount of maltotriose hydrolyzed to maltose and glucose. Further hydrolysis of maltose to glucose was not considered. In the absence of water (0%), methanol (5% *v/v*) was included in the reaction to provide a reactive nucleophile. In Figure 1, substrate conversion after 4 h of reaction is shown as a function of the water content. DES-containing acidic groups were not suitable solvents at any water concentration assayed, and were therefore not included in the figure.

As shown in Figure 1, significant α- amylase hydrolytic activity takes place in solvents containing only 10% water or less for several DES, except for choline chloride:malonic acid (1:1) and choline chloride:levulinic acid (1:2), for which no activity was observed. In order to include enzyme stability in the assay, samples were taken from the same reactions after 4 h and checked for residual amylase activity on starch. Figure 2 shows the residual amylase activity after 4 h reaction in DES, based on the I^-^/I_2_ staining assay. Residual amylose produces a blue stain, indicating that the amylase was deactivated by the corresponding DES at some time during the reaction. Clear wells indicate the presence of active amylase. The yellow color (column 7) was formed, probably from a side reaction of the DES with iodine.

In the last row of Figure 2, it may be observed that the enzyme was fully inactive after 4 h of reaction when 100% DESs were used as solvent. Also, the enzyme inactivated in those reactions containing acidic DESs (columns 3–4), probably as a result of a severe pH shift, as reported for acidic DES [32,33]. A similar inactivation profile was observed with DES 9 (choline chloride:benzyl alcohol (1:2)). In the case of choline chloride:urea (1:2) and choline chloride:1,3-propanediol:water (1:1:1), the activity was retained with 90% of DES. No information could be obtained on amylase stability in methyl (triphenyl) phosphoniumbromid (MPh_3_PBr):glycerol (column 7), probably due to a side reaction of the iodine-staining solution with the DES (yellow color). Amylase activity was retained in all reactions containing choline chloride:glucose:water (column 11) and choline chloride:sucrose:water (column 12). From this analysis, choline chloride:urea (1:2), choline chloride:1,3-propanediol:water (1:1:1), choline chloride:glucose:water (4:1:4); and choline chloride:sucrose:water (5:2:5) were defined as the most adequate DESs for amylase reactions. However, this is a preliminary qualitative analysis based on a staining procedure so further exploration of the enzyme stability is required. 

### 2.2. Screening for Transfer to Alcohols (Alcoholysis) in Selected DES

The initial screening evaluating the hydrolytic activity of the α-amylase in DES/water systems allowed us to identify DESs in which the α-amylase activity could be retained after several hours. The six most promising DESs were selected for a second screening with methanol as the acceptor to screen for alcoholytic activity (see materials and methods). As shown in Figure 1, water was required for Amy A stability, but as water also competes as a nucleophile, both water and methanol concentrations were varied in order to identify the conditions for an optimal alcoholysis:hydrolysis ratio. 

Methanol was assayed in a 5–30% *v/v* of total volume concentration range, with the remaining volume corresponding to the DES:water solvent. In the latter, water concentration ranged between 5 to 40% (*v/v*), while DES completed the remaining 60–95% (*v/v)*. The same stability assay where blue wells indicate enzyme deactivation was performed. A general trend may be observed in Figure 3 for all the assayed DESs, as low water and high methanol concentrations tend to deactivate the enzyme. In effect, when the assay was carried out in a DES:buffer solvent containing 90% DES (10% buffer), no activity was observed in all cases except for choline chloride:sucrose:water with low methanol concentrations of 5 or 10% (*v/v*). All other solvents required a minimum of 20% (*v/v*) water as a cosolvent. Although one would expect Amy A to tolerate low concentrations of water, the presence of water in some DES, like choline chloride:1, 3-propanediol:water (1:1:1) required an additional 10% more buffer in the DES:water solvent, as well as the choline chloride:glucose:water (4:1:4) DES that required 60%.

The alcoholysis reactions were analyzed by HPLC using an ELSD detector. However, the high concentration of the DESs components in some cases interfered with certain product peaks. Additionally, some DESs contained alcohols such as 1, 3-propanediol, glycerol, sucrose, and D-isosorbide. These alcohols represent potential nucleophiles in addition to methanol or water that could compete as acceptors in the transfer reactions. Accordingly, the unidentified peaks probably correspond to products formed from such alcoholysis reactions. However, reference compounds for these substances are not available, making identification and quantification difficult. From these results, we decided to select chloride:urea (1:2) DES as cosolvent. In the first experiment, maltotriose was used as the glycosyl donor. The results are summarized in Figure 4a–c. Conversion was based on the remaining maltotriose concentration after reaction.

Figure 4a shows that higher DES concentrations in the reaction result in lower conversion levels, most likely due to both low activity and stability of the enzyme (for stability see Figure 1 and Figure 3). Similarly, conversions levels also decrease in the presence of high methanol concentrations. It must be considered, however, that the conversion of maltotriose can be achieved by hydrolysis, alcoholysis, or transglycosylation. Therefore, in order to evaluate the specific production of the targeted methyl-glycosides, the concentrations of products obtained from alcoholysis were quantified by HPLC. Methyl-glucoside, methyl-maltoside, and interestingly, also methyl-maltotrioside were detected. The formation of the latter indicated a transfer reaction of oligosaccharides, as observed earlier by our group using Amy A [16]. In order to reduce the number of poly-glycosylated methanol products, a glucoamylase (GA) digestion treatment of the reaction products obtained with Amy A was carried out. It was found that this enzyme was capable not only of transforming the residual dextrins to glucose through the hydrolysis of glucose units linked to amylose non-reducing ends, but also of hydrolyzing the poly-glucosylated methanol forms to the simplest mono-glucosylated form [16,17]. The highest methyl-glycoside concentrations of approximately 39 mM (9.51 g/L) were obtained using methanol at 10% (*v/v*) in 60% choline chloride:urea (30% water). This concentration was higher than previously reported (7.5 g/L) [16]. Figure 4c shows the percentage of transfer to methanol as compared to hydrolysis. Higher methanol concentrations generally lead to a higher methanolysis vs. hydrolysis ratio, as expected. However, above 10% methanol (*v/v*), no further increase could be observed in methyl glycoside production in all reactions. The overall conversion was low with high DES concentrations (Figure 4a).

### 2.3. Exploration of Alcoholysis Activity in Choline Chloride:Urea (1:2):Water DES Using Starch as Glucosyl Donor

The use of maltotriose as a glycosyl donor showed promising results; however, this is an expensive substrate for industrial purposes. We therefore decided to use starch, as it is not only an economic alternative, but it also offers a stabilizing effect on amylases. In previous works, it was shown that with 15% (*w/v*) starch, conversions higher than 90% were obtained in methanol concentrations up to 20% (*v/v*) [16,18].

In a preliminary experiment, the alcoholysis reaction products using different concentrations of methanol (10, 20 and 30% *v/v*) and DES (0%–50% *v/v*) were quantified during the reaction in periods of 6, 12, and 24 h. Overall, the amount of methyl-glucoside (MG) obtained did not change much after 6 h of reaction (data not shown). DES produced an increment in alcoholysis yield when methanol was used at 10%; however, there was no further yield increase with higher concentrations of methanol in the DES system. In contrast, in a buffer system, higher alcoholysis yields were observed up to 30% methanol concentrations (data not shown). These and the results with maltotriose as a glycosyl donor prompted us to investigate whether DES could have a destabilizing effect on the enzyme, especially because the reaction was carried out at a high temperature (85 °C).

### 2.4. Stability of Amy A in Mixtures of DES-Methanol

The far-UV region of circular dichroism spectra is sensitive to changes in the secondary structure of proteins, and can be used to monitor the stability of the protein under different conditions. Figure 5 shows the CD spectra of the protein at DES concentrations from 60 to 80%. Light scattering due to the high DES concentration precluded the comparison of spectra below 210 nm. However, Figure 5 shows the negative signal around 220 nm, characteristic of proteins that present secondary structures. This negative signal looks similar for all the DES mixtures containing water at 25 °C, suggesting minimal changes in secondary structure. In contrast, Amy A in pure DES showed a near 50% decrease in the signal at this wavelength, suggesting the loss of the secondary structure, and in good agreement with the lack of activity observed under this condition. 

The activity assay, however, was carried out at 85 °C; therefore, we decided to carry out temperature scans of Amy A registering the signal at 220 nm at different concentrations of DES and MeOH. Figure 6 shows the change of signal by increasing and then decreasing the temperature from 25 °C to 90 °C; the insets show the CD spectra before and after heating to 90 °C and cooling back the protein solutions to 25 °C. Interestingly, the proteins in DES/water mixtures showed minimal differences from the proteins in the buffer, even though the concentrations of DES were as high as 80% (Figure 6a–c). The protein did not show significant changes in the mean residue ellipticity at 220 nm ([θ] mre) upon heating in any of the DES-buffer mixtures, showing a final spectrum that was practically unchanged. However, methanol turned out to be the major destabilizing factor. Even with only 10% methanol, a cooperative decrease in the signal was observed between 60 and 80 °C (Figure 6d–f). After reaching a minimum, the signal increased and the refolding process did not follow the same curve as the unfolding, showing considerable loss of the secondary structure. It is noteworthy that although the sample containing 80% DES lost about 35% of its [θ]mre at 220 nm, even at 25 °C (Figure 6f), after heating and cooling, it had a signal that was similar to those of the samples containing less DES, suggesting that the secondary structure was disturbed more by the increase of methanol concentration than by that of DES. When the methanol concentration increased to 20%, the sample containing 80% DES lost about 50% of the secondary structure, even at 25 °C (Figure 6i), but there was only a limited further loss of signal upon heating, and the refolding signal stayed steady. In contrast, the samples with 60% DES-20% MeOH and 70%DES-20% MeOH showed a high secondary structure content at 25 °C, and went through an irreversible unfolding when heated to 90 °C (Figure 6g,h). Again, the signal after cooling was similar for all samples containing 20% MeOH, around −1000 mdeg*cm^2^/dmol, implying a major dependence on MeOH concentration. 

Since major irreversible changes can occur in most of the conditions at temperatures above 60 °C, we decided to check the reversibility of heating the protein to only 60 °C for all conditions. As can be observed in Figure 7, even when there were subtle changes in [θ]mre at 220 nm upon heating, the final protein spectrum looked very similar to the initial one. Thus, we established 60 °C as the limit temperature to be able to compare the yield of alcoholysis reaction for all conditions. 

### 2.5. Activity of Amy A at 60 °C 

Amy A is a thermostable enzyme with an optimal temperature at 85 °C, an advantage under the high temperature required for starch solubilization and for DES viscosity reduction. However, the structural information we obtained established a temperature limit for the alcoholysis reaction which was well below the optimal temperature. We measured activity at 60 °C under the different solvent conditions, including with an aqueous buffer, which is not reported at this temperature. Table 1 shows the V_max_ and K_M_ values at 60°C for the hydrolysis reaction. 

Hydrolytic activity was reduced almost three-fold by decreasing the temperature to 60 °C, relative to that at T_opt_ (85 °C) in aqueous buffer in good agreement with the observations by Liebl, et al. [32]. At 60 °C, in the absence of DES, Amy A follows a typical M-M model (Appendix A). However, in the presence of 60% DES with or without 10% methanol, an inhibitory effect was observed as the starch concentration increased (Appendix A). The apparent inhibition might be explained by the high viscosity in the resultant reaction medium as a result of the high concentration of both DES and starch, making diffusion a limiting factor for the catalytic efficiency as observed in other enzymes by the addition of trehalose [34,35,36]. For this reason, initial velocities up to 8 mg/mL were fit to the Michelis-Menten equation to estimate catalytic parameters (Appendix A). Table 1 shows that hydrolytic activity decreases by almost an order of magnitude in the presence of DES, and that the addition of 10% MeOH decreases this value by another order of magnitude, while K_M_ increased twice in DES, reflecting the lower availability of substrate in the active site. The presence of methanol in DES, on the other hand, decreased slightly K_M_, probably as a result of reduced viscosity. Nevertheless, relevant for our purposes is the ratio of alcoholysis to hydrolysis reaction, which defines the final product yield and purity.

### 2.6. Alcoholysis Yield by Amy A at 60 °C

Alcoholysis initial rates are difficult to measure, since product quantification requires the hydrolysis of methyl-oligosaccharides and quantification of the resultant methyl-glucoside by HPLC (Appendix A). Instead, as a point of comparison, we decided to measure the final yield of both hydrolysis (measured by reducing sugars) and alcoholysis (measured by the amount of methyl-glucoside) after 18 h reaction, when presumably both reactions had reached equilibrium. We carried out alcoholysis with *T. maritima* α-amylase using 10% of methanol and varying the DES concentration from 10 to 80%. The product concentration after 18 h of incubation at 60 °C is compared to that obtained under the same conditions in the absence of DES in Figure 8. The systems with higher DES concentration (70 and 80%) have a very high viscosity when mixed with starch, making the medium and the measurement unmanageable. We excluded these experiments from the plots.

As shown in Figure 8, alcoholysis yield remained practically unchanged up to 20% DES, and still reasonably high up to 40% DES, while hydrolysis yield showed a constant decrease as DES concentration increased, so that the alcoholysis:hydrolysis ratio increased with the increase of DES concentration. 

## 3. Discussion

Water activity is a key parameter affecting glycosidase-catalyzed synthesis. On one hand, a sufficiently high-water activity (aw  =  0.6) is required by glycosidases to retain activity, but on the other hand, a high-water content promotes hydrolysis, thus decreasing the overall transfer reaction to alcohols [4]. In each catalytic event an α-amylase-glycan intermediate is formed whose sugar moiety may be transferred to water or methanol. The selectivity of this transfer depends on the concentration of nucleophiles and their specificities defined by their corresponding rate constants [27,30]. For most α-amylases, transglycosylation reactions are unfavorable under normal conditions. To become significant, they require a high concentration of substrates and low water activities [3,4]. Amy A has an important transglycosidic activity; however, its α-1,4 hydrolytic activity seems to be also higher, so a rather limited transference of glycosyl residues to methanol occurs [16,20]. Although DES was envisioned as a way to reduce water content in the reaction mixture, water is always required in small amounts to ensure activity and stability. Thus, achieving an anhydrous medium that would eliminate the hydrolysis reaction that competes with the production of alkyl glycosides is not possible, indicating that some hydration is needed to form an aqueous shell around the protein that allows it to fold and the necessary lubrication in the active site to work, as has also been observed by others [33]; this contrasts with lipases for which DES seem sufficient to mimic the water layer covering active enzymes in alternative reaction media to water [21,22]. 

Despite the need for water in the reaction medium, further studies with the choline chloride:urea DES demonstrated that DES can improve the reaction selectivity to the desired production of alkyl glucosides. Increasing concentrations of DES reduced both hydrolysis and alcoholysis, but this reduction was higher for the undesired hydrolysis reaction, reducing its production more than that of the alcoholysis reaction, resulting in an increase in selectivity.

Circular dichroism (CD) structural studies revealed that the loss of activity might result from thermal denaturation in reactions media with high DES content. Surprisingly, MeOH disturbed the structure of Amy A more than DES. This loss of structure was prevented by performing the reaction at temperatures below the transition temperature observed at the thermal denaturation curves. This decrease in temperature reduced the enzyme activity; as a result, the reaction time should be elongated to increase the yield of methyl glycosides.

## 4. Materials and Methods 

### 4.1. Materials 

Glucoamylase from *A<.spergillus niger*, choline chloride, soluble starch, maltotriose, urea, methyl urea, malonic acid, levulinic acid, ethylene glycol, D-isosorbide, MPh_3_PBr, glycerol, benzyl alcohol, 1,3-propanediol, glucose, sucrose, *α*-methyl-glucoside, and methanol were purchased from Sigma Chemical Co. (St. Louis, MO, USA). 

### 4.2. Analytical Procedures

Amylase from *T. maritima,* Amy A, was obtained from the heterologous expression in *E. coli* as described previously [16,20]. Cells containing the overexpressed protein were lysed by sonication and partially purified via heat precipitation. The protein concentration of the applied α-amylase stock was 15 mg mL^−1^ for initial screening. To determine catalytic parameters, the protein extracts were further purified through Ni^2+^ affinity columns as previously described [16]

### 4.3. Synthesis of Deep Eutectic Solvents (DES)

The DES used in this work were synthesized in two different ways. In the first method, a thermal approach was used, which consisted in mixing and heating the components of eutectic solvent until a liquid was formed. A vacuum was applied in order to avoid water entering the system, particularly in cases where hygroscopic compounds such as choline chloride were used. The second method consisted of the dissolution of the components in water followed by freeze-drying [37]. Appendix A, summarizes the preparation of each DES used in this work. 

### 4.4. Screening of Deep Eutectic Solvents for Application in α-amylase Catalyzed Reactions

Twelve DESs were used from the list presented in Appendix A (see Appendix A) for analysis in α-amylase-catalyzed reactions: 1. choline chloride:urea (1:2); 2. choline chloride:methyl urea (1:2), 3. choline chloride:malonic acid (1:1), 4. choline chloride:levulinic acid (1:2), 5. choline chloride:ethyleneglycol (1:2), 6. choline chloride:D-isosorbide (1:2); 7. MPh_3_PBr:glycerol (1:2), 8. choline chloride:glycerol (1:3); 9. choline chloride:benzyl alcohol (1:2); 10. choline chloride:1,3-propanediol:water (1:1:1); 11. choline chloride:glucose:H**_2_**O (4:1:4); 12. choline chloride:sucrose:H**_2_**O (5:2:5). The 12 DES were tested in 96-well PCR plates each at different water concentrations of 0, 5, 10, 25, 50, 75, and 100% (*v/v*) to determine the minimum amount of water needed for enzyme activity, using 50 mM maltotriose as substrate. In order to provide a reactive nucleophile, methanol (5% *v/v*) was added to those reactions not containing water (0%). The reaction containing 0% water contained methanol as a nucleophile instead of water, and lyophilized Amy A was used. The PCR plate was sealed with a silicone microplate sealing mat to avoid water or methanol evaporation. After 4 h incubation in a PCR cycler at 75 °C, samples of each reaction were quenched (20 µl of sample + 180 µl of 0.225 M HCl) and analyzed via HPLC to determine remaining maltotriose. 

### 4.5. Stability of Amy A in the Different Deep Eutectic Solvents

In order to determine the stability of Amy A during the reaction time that took place in the different DES systems, 20 µl samples from each reaction after 4 h incubation was diluted in 180 µl of a 50 mM Tris-HCl, 2 mM CaCl_2_ buffer at pH 7.0, containing 1% (*w/v*) starch and incubated at 50 °C for 1 h. To visualize the remaining enzyme activity, 100 µl of this reaction were diluted 1:2 in water and stained with 20 µl of a 5 mM I^-^/I_2_ solution.

### 4.6. Screening for Alcoholytic Activity in Selected DES 

Based on the first screening, only six different DES were further screened for alcoholytic activity: choline chloride:urea (1:2); choline chloride:D-isosorbide (1:2); choline chloride:glycerol (1:3); choline chloride:1,3-propanediol:water (1:1:1); choline chloride:glucose:water (4:1:4); choline chloride:sucrose:water (5:2:5). Methanol was tested in a range between 5–30% *v/v* of the total volume. The remaining volume was completed by a DES:buffer solution in which DES concentration ranged between 60–95% *v/v*. Thus, the six DES under the 16 different conditions could be evaluated in a 96-well plate. Then, 50 mM maltotriose was used as substrate, 5 µl of a 15 mg/mL Amy A solution obtained after purification by heat precipitation were added as described above. The total reaction volume was 150 µl in all reactions. Reactions were carried out at 75 °C in a PCR cycler. The same stability test as described above was carried out for each sample after 4 h of reaction. The reactions were additionally analyzed by HPLC. The percentage of conversion was estimated from the remaining maltotriose.

### 4.7. Product Quantification by HPLC

When maltotriose was used as substrate, the reaction products were analyzed by HPLC in an Ultimate 3000 equipped with a 300 RS Pump, a 3000 RS Autosampler and a RS Column Compartment, using an ELSD 2000ES Alltech detector. Both maltooligosaccharides and methyl-glucosides could be separated and detected using a Prevail Carbohydrate column (acetonitrile:water 70:30 as mobile phase, flow: 1 mL/min, column T: 30 °C, ELSD T: 80 °C, nitrogen Flow: 1.7 L/min). When starch was used as substrate, reaction products were analyzed in a Waters-Millipore 510 HPLC system equipped with an automatic sampler (model 717 Plus) (Waters Corp., MA, USA), a refractive-index detector (Waters 410) (Waters Corp., MA, USA) and a Hypersil GOLD™ Amino column (Thermo Scientific, Paisley, UK) using acetonitrile:water (80:20) as the mobile phase at a flow rate of 1.0 mL/min. The peak areas were measured and compared against those of a standard curve containing known amounts of methyl-glucoside.

To calculate the % conversion of the reaction, the following formula was used:
$$\%Conversion=\frac{{\left[Maltotriose\right]}_{inicial}-{\left[Maltotriose\right]}_{final}}{{\left[Maltotrios\right]}_{inicial}}\times 100$$

### 4.8. Hydrolysis Activity Assay

The enzyme activity of Amy A from *T. maritima* was determined by measuring the initial velocities of reducing sugars released at saturation starch concentration by the 3,5-dinitrosalicylic acid method, as reported previously [38]. The reaction was carried out in 1 mL of 10 mg/mL soluble starch dissolved in 50 mM Tris, 150 mM NaCl, 2 mM CaCl_2_ buffer, pH 7, at 85 °C. A unit of enzyme activity was defined as the amount of enzyme required to release 1 µmol of glucose equivalents per minute. For the characterization of Amy A at 60 °C, the activity measurements were carried out varying the starch concentration from 1 mg/mL to 20 mg/mL and using as a solvent mixtures containing 50 mM Tris, 150 mM NaCl, 2 mM CaCl_2_ buffer, pH 7, and variable *v/v* concentrations of DES and methanol. The initial velocities were plotted against starch concentration and the data fit to the Michaelis-Menten equation to obtain the catalytic parameters. 

### 4.9. Alcoholysis Reactions with Methanol and Starch in Choline Chloride:Urea 1:2 DES as co Solvent

Alcoholysis reactions were performed in 2 mL Eppendorf tubes which were tightly sealed to avoid methanol and water evaporation. DES concentration was varied from 0 to 80%, using 10% methanol and 10% starch as a glycosyl donor, and 50 mM Tris, 150 mM NaCl, 2 mM CaCl_2_ buffer at pH 7 to complete the 100% volume. The solvent mixture was homogenized and pre-equilibrated at the reaction assay temperature before adding 20 U of Amy A to start the reaction, which was incubated during 18 h at the defined temperature according to the experiment. 

### 4.10. Hydrolysis Quantification at the End Point of Reaction

The determination of reducing sugars was evaluated as described above from a 1:50 dilution of the final reaction. The hydrolysis events were estimated by subtracting the equivalents of the alcoholysis reaction from the equivalent of the reducing sugars. 

### 4.11. Alcoholysis Reaction Measurement

The measurement of alcoholysis is complicated by the diversity of products possible. To reduce the complexity, the final reaction products were hydrolyzed for 12 h at 50 °C with *A. niger* glucoamylase (25 U/mL), which is an exoglycosidase that hydrolyses α-1,4-glycosidic bonds from the non-reducing end, transforming any alcoholysis product in glucose plus methyl-glucoside for each alcoholysis event. At the end of incubation period, all the samples were centrifuged at 13,000 rpm for 15 minutes and filtered through a 0.22 µm nylon membrane before the analyses by HPLC as described above, using a standard curve of methyl-glucoside to quantify the amount in each sample.

### 4.12. Alcoholysis Yield and Alcoholysis/Hydrolysis Ratio

When a hydrolysis reaction takes place, for each molecule of a reducing substrate cleaved, one additional reducing end is formed, increasing in one its contribution to the solution reducing power; in contrast, when an alcoholysis reaction occurs, there is no real gain in reducing power. Therefore, a measure of the increasing equivalent glucose concentration measured by the DNS method, as described above, reflects the number of hydrolysis events that occurred during the reaction, while the number of alcoholysis events can be independently determined through the quantification of the increase in alkyl glucoside concentration in the same period of time. From these data it is possible to determine the number of hydrolysis and alcoholysis events in the reaction, as well as an alcoholysis/hydrolysis ratio and an alcoholysis yield:
$$\mathrm{Alcoholysis}/\mathrm{hydrolysis}\;\mathrm{ratio}=\frac{\mathrm{AlcoholysisEvents}}{\mathrm{HydrolysisEvents}}$$
$$\mathrm{Alcoholysis}\;\mathrm{efficiency}=\frac{\mathrm{AlcoholysisEvents}}{\mathrm{AlcoholysisEvents}+\mathrm{HydrolysisEvents}}$$

### 4.13. Structural Characterization of Amy A in Choline Chloride:Urea 1:2 DES as co Solvent

Circular dichroism spectra (CD) were recorded on a J-710 spectropolarimeterTM (Jasco, USA) equipped with a Peltier temperature control using a 1.00 mm path length cell. Three CD scans were recorded and averaged from 190 to 260 nm for each sample containing 0.3 mg/mL of Amy A in 10 mM Phosphate, 0.5 mM CaCl_2_ buffer at pH 7 at 25 °C and varying the DES (60%–80% *v/v*) and methanol (0, 10 and 20 % (*v/v*)) concentrations.

### 4.14. Thermal Stability of Amy A in Choline Chloride:Urea 1:2 DES as co Solvent

Thermal stability of Amy A was determined by following the CD signal at 220 nm during a temperature scan from 25 °C to 90° at a rate of 1 °C/min of all the samples described above. The reversibility of any change observed was also evaluated by following the CD signal at 220 nm in the return to 25 °C and recording a final spectrum for comparison with the initial one.

## 5. Conclusions

We demonstrated that Amy A can perform both hydrolysis and alcoholysis in media containing deep eutectic solvents (DES) using starch or maltotriose as substrates. However, the enzyme activity and stability in each medium depends on the nature of the DES. The possibility of reducing water activity using alternative reaction media as the DES cosolvent was successful to a certain degree, as shown by the increase in the alcoholysis/hydrolysis ratio; however, it was found that using pure DES had a destabilizing effect. The alcholysis yield measured as methyl-glucoside with the optimal DES cosolvent was 20% greater than that previously reported in an aqueous buffer with the same methanol concentration.

Our results highlight the importance of performing variable temperature structural studies to determine the temperature limit at which an enzyme is operative.

## Figures and Tables

**Figure 1 ijms-20-05439-f001:**
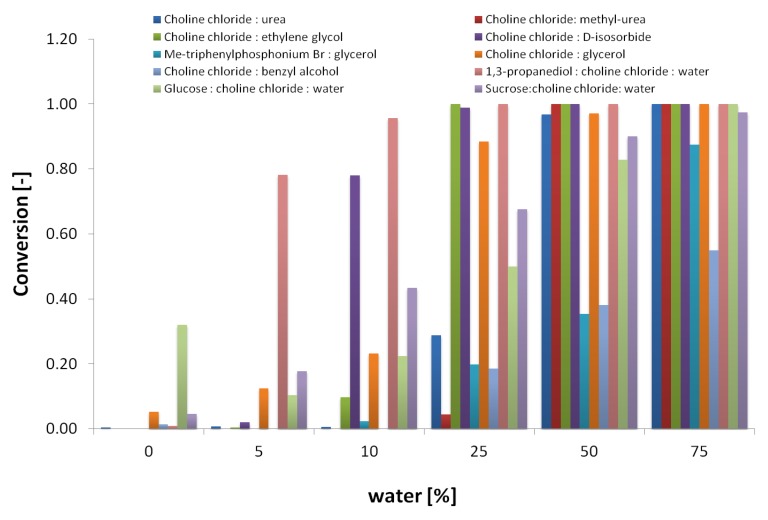
Effect of water content in water:DES cosolvent systems on maltotriose conversion in α- amylase-catalyzed hydrolysis reactions. No reaction was observed on choline chloride:malonic acid (1:1) and choline chloride:levulinic acid (1:2).

**Figure 2 ijms-20-05439-f002:**
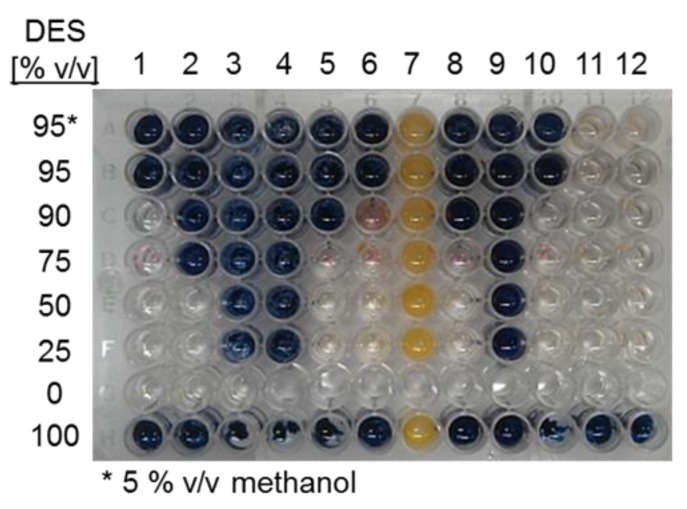
Screening for α-amylase stability in DES:water cosolvent mixtures as detected by the I^-^/I_2_ staining assay with amylose, after 4h reactions in the corresponding DES. **1**. choline chloride:urea (1:2); **2**. choline chloride:methyl urea (1:2), **3**. choline chloride:malonic acid (1:1), **4**. choline chloride:levulinic acid (1:2), **5**. choline chloride:ethylene glycol (1:2), **6**. choline chloride:D-isosorbide (1:2); **7**. MPh_3_PBr:glycerol (1:2), **8**. choline chloride:glycerol (1:3); **9**. choline chloride:benzyl alcohol (1:2); **10**. choline chloride:1,3-propanediol:water (1:1:1); **11**. choline chloride:glucose:water (4:1:4); **12**. choline chloride:sucrose:water (5:2:5).

**Figure 3 ijms-20-05439-f003:**
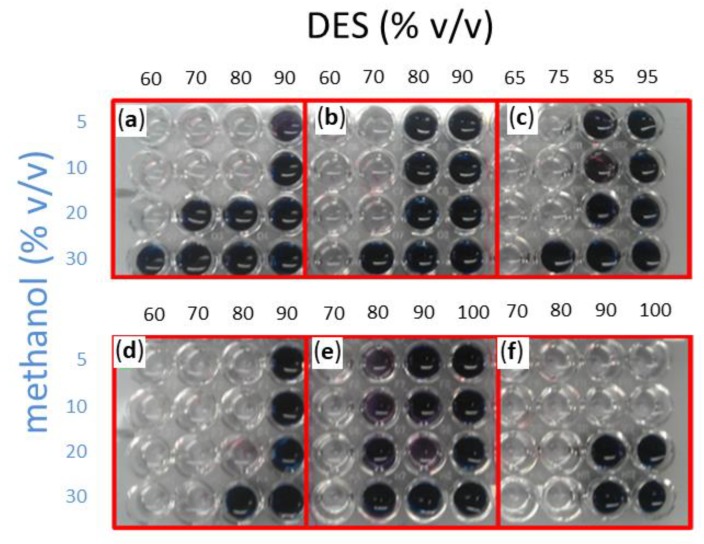
Retained Amy A activity as evaluated after 12 h of reaction in DES:water:MeOH systems at 75 °C. A reaction aliquot was taken to see retaining activity. The assay shown here was carried out with 1% starch (*w/v*) as a substrate at 50 °C. Six different DESs were screened: (**a**) choline chloride:urea (1:2); (**b**) choline chloride:D-isosorbide (1:2); (**c**) choline chloride:glycerol (1:3); (**d**) choline chloride:1,3-propanediol:water (1:1:1); (**e**) choline chloride:glucose:water (4:1:4); (**f**) choline chloride:sucrose:water (5:2:5). Methanol was tested in a range between 5–30% *(v/v)* of the total volume as indicated. The remaining volume was completed by a DES:buffer solution in which DES concentration was varied as indicated in each panel.

**Figure 4 ijms-20-05439-f004:**
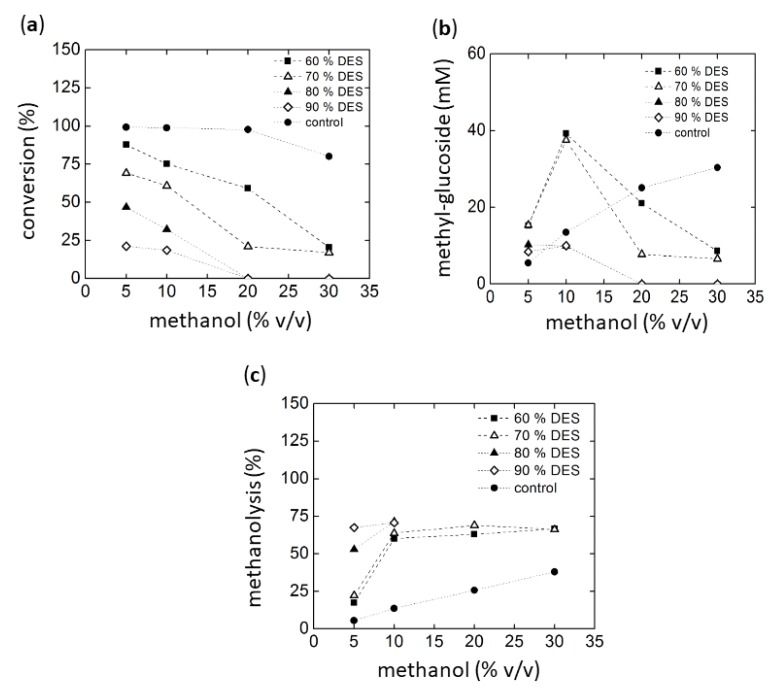
Methanolysis with maltotriose as substrate using Amy A as catalyst in choline chloride:urea (1:2):water as a solvent with increasing amounts of MeOH. **(a**) Maltotriose conversion**, (b**) Alcoholysis evaluated as methyl-glucoside synthesis after glucoamylase treatment of the glucosylated products (see text), (**c**) final yield of methanolysis reaction.

**Figure 5 ijms-20-05439-f005:**
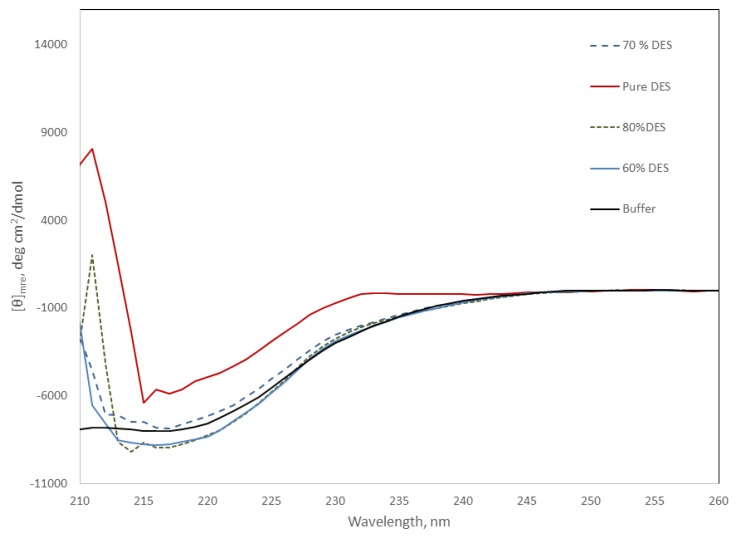
Far-UV region of the CD spectra of Amy A in pure phosphate buffer and in choline chloride:urea (2:1) DES/buffer mixtures (20 to 100% (*v/v*) of DES) at 25 °C. Pure DES solid red line; 80% DES dashed green line; 70% DES blue dashed line; 60% DES blue solid line; buffer, black solid line.

**Figure 6 ijms-20-05439-f006:**
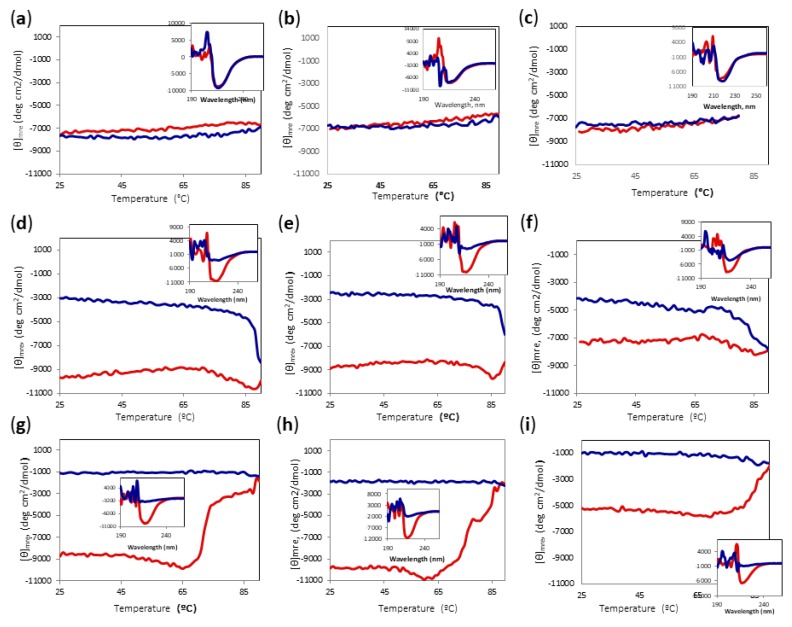
CD temperature scans of Amy A from 25 °C to 90 °C in different choline chloride:urea (1:2) DES and MeOH concentrations. Main chart: Amy A heating signal (red) and cooling signal (blue) at 220 nm. Inset: Far UV-CD spectra before heating (red) and after cooling (blue). (**a**) 60% DES; (**b**) 70% DES; (**c**) 80% DES; (**d**) 60% DES, 10% MeOH; (**e**) 70% DES, 10% MeOH; (**f**) 80% DES, 10% MeOH; (**g**) 60% DES, 20% MeOH; (**h**) 70% DES, 20% MeOH; (**i**) 80% DES, 20% MeOH.

**Figure 7 ijms-20-05439-f007:**
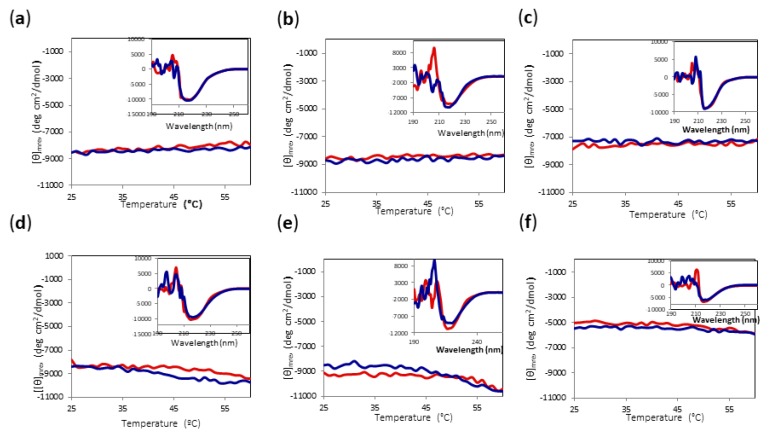
CD temperature scans of Amy A from 25 °C to 60 °C in different choline chloride:urea (1:2) DES and MeOH concentrations. Main chart: Amy A heating signal (red) and cooling signal (blue) at 220 nm. Inset: Far UV-CD spectra before heating (red) and after cooling (blue). (**a**) 60% DES, 10% MeOH; (**b**) 70% DES, 10% MeOH; (**c**) 80% DES, 10% MeOH; (**d**) 60% DES, 20% MeOH; (**e**) 70% DES, 20% MeOH; (**f**) 80% DES, 20% MeOH.

**Figure 8 ijms-20-05439-f008:**
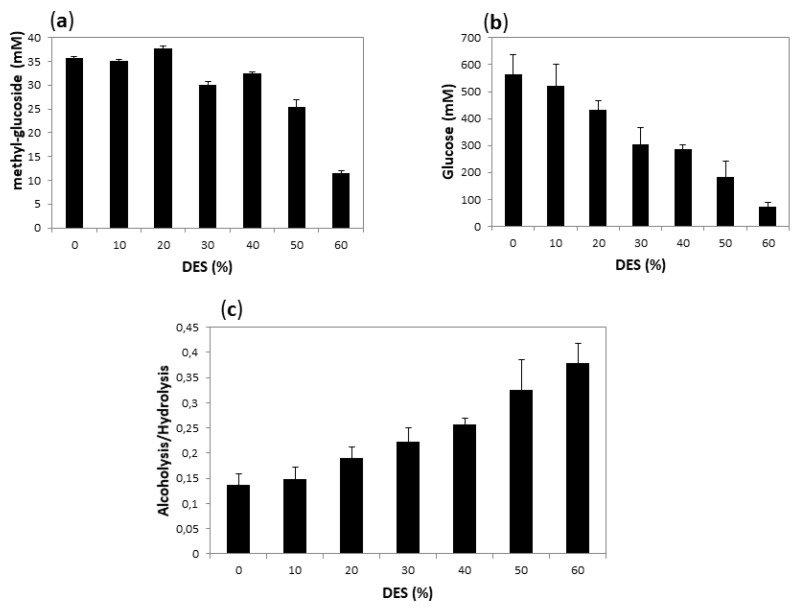
Methanolysis of starch after 18 h using Amy A as a catalyst, and 10% MeOH in different concentrations of choline chloride:urea (1:2):water DES as a solvent. (**a**) shows the yield of alcoholysis reaction (expressed as mM of methyl-glucoside); (**b**) the yield of hydrolysis (expressed as equivalent to dextrose in mM) and (**c**) the alcoholysis/hydrolysis ratio. The values were calculated as described in Materials and Methods.

**Table 1 ijms-20-05439-t001:** Kinetic constants of amylase Amy A.

Condition	K_M_ (mg/mL)	V_max_ (µmol/mg *min)
No DES	2.8 ± 0.5	196 ± 10
60% DES	5.7± 1.7	27 ± 4
60% DES 10% MeOH	1.7± 0.6	3.6 ± 0.4

Measurements at 60 °C, in buffer 50 mM Tris pH7 containing 150 mM NaCl and 2 mM CaCl_2_. The results are the average of three replicates and the errors represent one standard deviation.

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
