# Peer review of "Deep Eutectic Solvents as New Reaction Media to Produce Alkyl-Glycosides Using Alpha-Amylase from Thermotoga maritima"

_ijms, 2019, doi:10.3390/ijms20215439_

Round 1

Reviewer 1 Report

I'm convinced that the research presented in the manuscript is very interesting and worth to publish. I recommend it for publication almost as it is. Only some minor corrections are needed.

There are some misspells in the text, like hours at some places are abbreviated with capital H, and at some places milliliters are abbreviated by ml instead of mL. 

In the reference section or more precise the beginning of the section seems to be remains from the templates instructions, those have to be deleted. In many references journal full names are given instead of abbreviations. And finally I could not find reference 34 cited anywhere in the text. Please check it and correct if needed.

After corrections are done, I recommend the manuscript for publication without additional review. 

Author Response

We want to thank the time and comments of the reviewers, that we are sure, will make a better presentation of the work.

Point by point answer to reviewer 1.

There are some misspells in the text, like hours at some places are abbreviated with capital H, and at some places milliliters are abbreviated by ml instead of mL.

R: abbreviation of hours has been replaced by “h” in all cases and milliliters “ml” by “mL”.

In the reference section or more precise the beginning of the section seems to be remains from the templates instructions, those have to be deleted.

R: Thank you for the observation. These lines from the template instructions have been removed.

In many references journal full names are given instead of abbreviations.

R: The journal full names have been replaced by their abbreviations

I could not find reference 34 cited anywhere in the text. Please check it and correct if needed.

R: This was checked and corrected. Some references have been added, so that now that reference is number 38 and is mentioned in the text M&M section in the Hydrolysis activity assay section.

Reviewer 2 Report

Comments

Line 18, "reactions" instead of "reations"

Line 83 “Tabla” should be “Table”

Line 127, the authors suggest that inactivation “probably caused by a severe pH shift”, did the authors check this?  Could they add pKa of the DES involved?

Line 225, the authors refer a “characteristic peak around 220 nm looks similar for all the DES”, from the figure provided it is not evident which peak they refer to. Could the authors highlight the peak in the graphic?

Line 279, the authors state “shows the kcat and Km values”, yet Table 1 provides data for Km and Vmax. Although Vmax and kcat are related, they are not the same, please correct

Additionally in Table 1, data for kinetic parameters in the presence of both DES and DES+MetOH are not statistically meaningful, as the error/standard deviation associated are far higher than the average value, leading to confidence intervals that can be about 2- to about 4-fold the average. Were these obtained when the product inhibition model was used (and which model was this)? How many replicates were performed to establish the kinetic parameters? Could the authors add error bars for the data presented in Fig S1? Please comment on this?

Line 288, the authors refer to “Figure S1b” yet there in only Figure S1 (where the trend for inhibition at high substrate concentration in the presence of DES is noticed), please check

Lines 288/289, the authors suggest that “The apparent inhibition might be explained by the high viscosity in the resultant reaction medium”, please add reference form the literature to support this claim

Line 357 “Thermotoga maritima” must be in italic

Line 380/381 “The PCR plate was tightly closed” could the authors provide some information on how exactly the plates were tightly closed? Were membranes used? Other? 

Line 412, “80;20” should be “80:20”

Line 415, the authors refer specifically “wild-type Amy A”, which is not referred elsewhere. Still, could the authors confirm that Amylase AmyA from Thermotoga maritima expressed in E. coli was used in all experiments?

Lines 420/421 , the authors state that “For the characterization of Amy A at 60°C, the activity measurements were carried out varying the starch concentration from 1 mg/mL to 10 mg/mL”, but in Figure S1 starch concentration reported is up to 20 mg/mL, please check. Additionally, and although the authors provide the units of the reaction rate in the legend, could they also add this information in the y axis? This would be consistent with the information in x axis, where concentration units are provided in both said axis and legend.

Line 440, there is one” with” too many 

Line 443, “At” rather than “AT”

Line 447, “Circular dichroism spectrum (CD) were recorded” should be “Circular dichroism spectra (CD) were recorded”

Author Response

Point by point answer to reviewer 2.

We want to thank the time and comments of the reviewers, that we are sure, will make a better presentation of the work.

Line 18, "reations" has been replaced by "reactions" Line 83 “Tabla” has been replaced by “Table” Line 127, the authors suggest that inactivation “probably caused by a severe pH shift”, did the authors check this? Could they add pKa of the DES involved?

R: We did not check it but Qin et al. and Skulcova et al. (references 32 and 33 now included) demonstrated that the acidity of DES was assessed by the acidity of the Hydrogen-Bond Donor (HBD) in DES, being more acidic those DES using more acidic HBD.  

Line 225, the authors refer a “characteristic peak around 220 nm looks similar for all the DES”, from the figure provided it is not evident which peak they refer to. Could the authors highlight the peak in the graphic?

R: A statement in lines 232-234 has been added, clarifying that having a negative signal around 220 nm (as in the spectra shown) is indicative of the formation of secondary structure.

Line 279, the authors state “shows the kcat and Km values”, yet Table 1 provides data for Km and Vmax. Although Vmax and kcat are related, they are not the same, please correct.

R: Thank you for noticing this error, this has been corrected in the text.

Additionally, in Table 1, data for kinetic parameters in the presence of both DES and DES+MetOH are not statistically meaningful, as the error/standard deviation associated are far higher than the average value, leading to confidence intervals that can be about 2- to about 4-fold the average. Were these obtained when the product inhibition model was used (and which model was this)? How many replicates were performed to establish the kinetic parameters? Could the authors add error bars for the data presented in Fig S1? Please comment on this?

R: In fact, we observe that at high substrate (starch) concentration, an apparent inhibitory effect is observed. However, as it has been demonstrated, viscosity has a negative effect on Vmax (or kcat) of enzymes (references 34-36 are included in the text). DES have high viscosity by themselves and the increment in starch concentration increases further the reaction medium viscosity. We therefore, eliminated data beyond 8 mg/mL of substrate to be able to apply the MM equation, since we attribute the loss of activity at high substrate concentration to the high viscosity, rather than to an inhibition phenomenon. This reduced the errors in the catalytic parameters estimation and an explanation is offered in the text (lines 301-304).

Line 288, the authors refer to “Figure S1b” yet there in only Figure S1 (where the trend for inhibition at high substrate concentration in the presence of DES is noticed), please check

R: This has been corrected and indeed an additional Figure (Figures S2) has been added to show the data of initial velocities in DES fit to Michaelis-Menten equation.

Lines 288/289, the authors suggest that “The apparent inhibition might be explained by the high viscosity in the resultant reaction medium”, please add reference form the literature to support this claim References 34-36 have been added to support this statement. Line 357 “Thermotoga maritima” must be in italic

R: This has been corrected. Thank you

Line 380/381 “The PCR plate was tightly closed” could the authors provide some information on how exactly the plates were tightly closed? Were membranes used? Other?

R: The plates were closed using

Line 412, “80;20” should be “80:20”

R: This has been corrected. Thank you

Line 415, the authors refer specifically “wild-type Amy A”, which is not referred elsewhere. Still, could the authors confirm that Amylase AmyA from Thermotoga maritima expressed in E. coli was used in all experiments?

R: We only worked with wild-type Amy A in all experiments, therefore, this specification was removed form line 415

Lines 420/421 , the authors state that “For the characterization of Amy A at 60°C, the activity measurements were carried out varying the starch concentration from 1 mg/mL to 10 mg/mL”, but in Figure S1 starch concentration reported is up to 20 mg/mL, please check. Additionally, and although the authors provide the units of the reaction rate in the legend, could they also add this information in the y axis? This would be consistent with the information in x axis, where concentration units are provided in both said axis and legend.

R: This has been corrected and clarified that for data analysis we only used concentrations below 10 mg/mL. The units have also been included in the Y-axis of the plot.

Line 440, there is one” with” too many

R: this was corrected. Thank you

Line 443, “At” rather than “AT”

R: This was corrected. Thank you

Line 447, “Circular dichroism spectrum (CD) were recorded” should be “Circular dichroism spectra (CD) were recorded”

R: “spectrum” was replaced by “spectra”. Thank you

Reviewer 3 Report

The manuscript “Deep eutectic solvents as new reaction media to 3 produce alkyl-glycosides”... is devoted to studies of the behavior and functionality of the bacterial enzyme alpha-amylase under specific conditions. The general aim of this work was to develop the conditions for the enzymatic synthesis of alkyl glycosides demanded for many applications. The idea to use deep eutectic solvents for this purpose is not radically new but is being used for the specific synthesis by alpha-amylase for the first time. One of the problems that the authors tried to solve was to establish the effect of DES on the stability and activity of the selected amylase.

It should be noted that the use of DES is rather original and new approach and, of course, an understanding of the limitations that appeare when using such substances in enzymatic synthesis is necessary.

I have few comments on the text.

1. The term "alcoholysis", which the authors use throughout the article, is not correct. Lysis means hydrolysis, cleavage of the bond. From the manuscript context is clear that transglycosylation products are obtained as a result of the enzymatic reaction both when the acceptor is a glycoside and/or any other hydroxyl-containing compound. It does not matter whether it is sugar, alcohol or amino acid. In any case, under certain conditions, transglycosylation can occur in reactions with retaining glycoside hydrolases. Authors should replace the word alcoholysis with transglycosylation throughout the text.

2. The manuscript presents mainly the qualitative results (Figs. 1-3). It is incluear how the data presented in Figures 4-8 were calculated (particularly, what formulae were used?).

3. Results: section 2.1 should be combined with 2.2 section since no results were presented there.

4. Table 1 shows the values of Vmax, not kcat, as described in the text.

5. In M&M section, HPLC separation of hydrolysis and transglycosylation products are descibed but the results of these experiments are not found in the Results section. They should be given at least in Supplementary materials.

6. As a concluding part, it would be nice to give quantitative estimates of the enzymatic synthesis of alkyl glycosides performed under the selected optimal conditions.

7. English style and numerous misprints should be corrected.

Author Response

We want to thank the time and comments of the reviewers, that we are sure, will make a better presentation of the work.

Point by point answer to reviewer 3.

The term "alcoholysis", which the authors use throughout the article, is not correct. Lysis means hydrolysis, cleavage of the bond. From the manuscript context is clear that

transglycosylation products are obtained as a result of the enzymatic reaction both when the acceptor is a glycoside and/or any other hydroxyl-containing compound. It does not

matter whether it is sugar, alcohol or amino acid. In any case, under certain conditions, transglycosylation can occur in reactions with retaining glycoside hydrolases. Authors should replace the word alcoholysis with transglycosylation throughout the text.

R: Just to clarify, lysis refers to cleavage, when this cleavage is carried out by a water molecule, then the reaction is hydrolysis, in whose case, one proton from the water molecule is added in one of the fragments and the hydroxyl group is added to the other fragment. In analogy, when an alcohol is acting as acceptor for the cleaved fragment, the reaction is named alcoholysis. In the field the reaction is named alcoholysis as reported by other authors:
Morrill et al., Applied Microbiology and Biotechnology (2018) 102:5149–5163

Moreno-Beltran, et al., Journal of Molecular Catalysis B: Enzymatic 6 (1999)1–10

Larsson, et al., Journal of Molecular Catalysis B: Enzymatic 37 (2005) 84–87

The manuscript presents mainly the qualitative results (Figs. 1-3). It is incluear how the data presented in Figures 4-8 were calculated (particularly, what formulae were used?).

R: Certainly, data shown in Figures 1-3 is qualitative. In the section of materials and methods, lines 1296, 1297 and 1329-1357 and explanation including the formulas used to calculate % conversion and transfer to alcohol/hydrolysis ratio have been included.

Results: section 2.1 should be combined with 2.2 section since no results were presented there.

R: The reviewer comment is very appropriate; these two sections have been consolidated in one.

Table 1 shows the values of Vmax, not kcat, as described in the text.

R: The reviewer is right, we have corrected the text.

In M&M section, HPLC separation of hydrolysis and transglycosylation products are described but the results of these experiments are not found in the Results section. They should be given at least in Supplementary materials.

R: The results are summarized as the alcoholyisis/hydrolysis ratio in figures 4 and 8. However a series of representative chromatograms have been included in Supplementary material. It is worth to clarify that, as described in M&M, hydrolysis was quantified through the determination of reducing power and not by HPLC.

As a concluding part, it would be nice to give quantitative estimates of the enzymatic synthesis of alkyl glycosides performed under the selected optimal conditions.

R: A quantitative estimation of the enzymatic synthesis of alkyl glycoside have been added in the conclusion section.

English style and numerous misprints should be corrected.

R: We have asked to proof read the manuscript to a more fluent colleague and improvements in the English language and style have been performed.